# Simple scoring tool to estimate risk of hospitalization and mortality in ambulatory and emergency department patients with COVID-19

**Brandon J. Webb**[1,2]*, **Nicholas M. Levin**[3], **Nancy Grisel**[4], **Samuel M. Brown**[5], **Ithan D. Peltan**[5], **Emily S. Spivak**[6], **Mark Shah**[7], **Eddie Stenehjem**[1,2,8], **Joseph Bledsoe**[7,9]

1 Division of Infectious Diseases and Clinical Epidemiology, Intermountain Healthcare, Salt Lake City, UT, United States of America, 2 Division of Infectious Diseases and Geographic Medicine, Stanford Medicine, Palo Alto, CA, United States of America, 3 Division of Emergency Medicine, University of Utah School of Medicine, Salt Lake City, UT, United States of America, 4 Intermountain Healthcare, Enterprise Analytics, Salt Lake City, UT, United States of America, 5 Division of Pulmonary and Critical Care Medicine, Intermountain Medical Center and University of Utah, Salt Lake City, UT, United States of America, 6 Division of Infectious Diseases, University of Utah School of Medicine, Salt Lake City, UT, United States of America, 7 Intermountain Healthcare, Department of Emergency Medicine, Salt Lake City, UT, United States of America, 8 Intermountain Healthcare, Office of Patient Experience, Salt Lake City, UT, United States of America, 9 Stanford Medicine, Department of Emergency Medicine, Palo Alto, CA, United States of America

* Brandon.Webb@imail.org

## Abstract

### Background

Accurate methods of identifying patients with COVID-19 who are at high risk of poor outcomes has become especially important with the advent of limited-availability therapies such as monoclonal antibodies. Here we describe development and validation of a simple but accurate scoring tool to classify risk of hospitalization and mortality.

### Methods

All consecutive patients testing positive for SARS-CoV-2 from March 25-October 1, 2020 within the Intermountain Healthcare system were included. The cohort was randomly divided into 70% derivation and 30% validation cohorts. A multivariable logistic regression model was fitted for 14-day hospitalization. The optimal model was then adapted to a simple, probabilistic score and applied to the validation cohort and evaluated for prediction of hospitalization and 28-day mortality.

### Results

22,816 patients were included; mean age was 40 years, 50.1% were female and 44% identified as non-white race or Hispanic/Latinx ethnicity. 6.2% required hospitalization and 0.4% died. Criteria in the simple model included: age (0.5 points per decade); high-risk comorbidities (2 points each): diabetes mellitus, severe immunocompromised status and obesity (body mass index≥30); non-white race/Hispanic or Latinx ethnicity (2 points), and 1 point

**Data Availability Statement:** In order to protect patient privacy and comply with institutional data use policy, data used in this study are unavailable to upload to public servers. As required by the

Intermountain Healthcare Institutional Review Board, data sharing agreement requests to access deidentified versions of the datasets generated and/ or analyzed during the current study may be addressed to the Intermountain Office of Research (officeofresearch@imail.org).

**Funding:** IP reports salary support through a grant from the National Institutes of Health (U.S.A). SB reports salary support from the U.S. NIH, Centers for Disease Control and the Department of Defense; he also reports receiving support for chairing a data and safety monitoring board for a respiratory failure trial sponsored by Hamilton, effort paid to Intermountain for steering committee work for Faron Pharmaceuticals and Sedana Pharmaceuticals for ARDS work, support from Janssen for Influenza research, and royalties for books on religion and ethics from Oxford University Press/Brigham Young University. BW reports partial salary support from a U.S. Federal grant from AHRQ. ES receives partial salary support through grants from the Centers for Disease Control. At the time of submission, Intermountain Healthcare and the University of Utah have participated in COVID-19 trials sponsored by: Abbvie, Genentech, Gilead, Regeneron, Roche, and the U.S. National Institutes of Health ACTIV and PETAL clinical trials networks; several authors (BW, IP, JB, SB, ES) were site investigators on these trials but received no direct or indirect remuneration for their effort. ES, BJW, SMB and MS are members of the Utah crisis standards of care scarce medication committee.

**Competing interests:** IP reports salary support through a grant from the National Institutes of Health (U.S.A). SB reports salary support from the U.S. NIH, Centers for Disease Control and the Department of Defense; he also reports receiving support for chairing a data and safety monitoring board for a respiratory failure trial sponsored by Hamilton, effort paid to Intermountain for steering committee work for Faron Pharmaceuticals and Sedana Pharmaceuticals for ARDS work, support from Janssen for Influenza research, and royalties for books on religion and ethics from Oxford University Press/Brigham Young University. BW reports partial salary support from a U.S. Federal grant from AHRQ. ES receives partial salary support through grants from the Centers for Disease Control. At the time of submission, Intermountain Healthcare and the University of Utah have participated in COVID-19 trials sponsored by: Abbvie, Genentech, Gilead, Regeneron, Roche, and the U.S. National Institutes of Health ACTIV and PETAL clinical trials networks; several authors (BW, IP, JB, SB, ES) were site

each for: male sex, dyspnea, hypertension, coronary artery disease, cardiac arrythmia, congestive heart failure, chronic kidney disease, chronic pulmonary disease, chronic liver disease, cerebrovascular disease, and chronic neurologic disease. In the derivation cohort (n = 16,030) area under the receiver-operator characteristic curve (AUROC) was 0.82 (95% CI 0.81–0.84) for hospitalization and 0.91 (0.83–0.94) for 28-day mortality; in the validation cohort (n = 6,786) AUROC for hospitalization was 0.8 (CI 0.78–0.82) and for mortality 0.8 (CI 0.69–0.9).

## Conclusion

A prediction score based on widely available patient attributes accurately risk stratifies patients with COVID-19 at the time of testing. Applications include patient selection for therapies targeted at preventing disease progression in non-hospitalized patients, including monoclonal antibodies. External validation in independent healthcare environments is needed.

## Introduction

COVID-19 is a systemic infection caused by a novel betacoronavirus, SARS-COV-2 [1]. A relatively conserved set of clinical and demographic factors are now recognized to correlate with an increased risk for severe disease requiring hospitalization, mechanical ventilation and death [2–4]. Accurate methods of risk stratifying ambulatory patients at the point of test positivity has many possible applications, including prioritizing patients at highest risk of hospitalization for early treatments aimed to prevent progression to severe disease, such as monoclonal antibodies, which are both limited in availability and also more likely to be effective in high-risk groups. Several models have been proposed [3,5–9]. We describe development and validation of a simple scoring model to predict hospitalization and mortality in a large cohort of ED and ambulatory patients with COVID-19.

## Methods

Intermountain Healthcare is an integrated healthcare system that provides care to more than 1.5 million patients each year in Utah and bordering communities. As part of a systemwide COVID-19 response, Intermountain provides SARS-CoV-2 testing at 32 urgent care facilities, 23 emergency departments, and 16 community drive-up testing sites. During the study period, only polymerase chain reaction (PCR) assays were performed (Thermofisher, Waltham, MA; Cepheid, Sunnyvale, CA; Quidel, San Diego, CA, BioFire, Salt Lake City, UT; Roche, Basel, Switzerland). All testing required an order entered in the electronic health record (EHR) (Cerner, Kansas City, KS) by the ordering clinician through a structured form that requires the clinician to input the patient's clinical symptoms and epidemiological features. These data are stored in the Intermountain Prospective Observational COVID-19 (IPOC) database, and the enterprise data warehouse.

We queried the IPOC database for consecutive adult patients with positive SARS-CoV-2 tests from March 25-October 1, 2020. Symptom data were extracted from the electronic test order form while demographic and co-morbidity data were obtained from the IPOC database and data warehouse using the Charlson and Elixhauser definitions [10,11]. We defined immunosuppression as: recipient of a solid organ or hematopoietic stem cell transplant, on

investigators on these trials but received no direct or indirect remuneration for their effort. ES, BJW, SMB and MS are members of the Utah crisis standards of care scarce medication committee. This does not alter our adherence to PLOS ONE policies on sharing data and materials.At the time of submission, Intermountain Healthcare and the University of Utah have participated in COVID-19 trials sponsored by: Abbvie, Genentech, Gilead, Regeneron, Roche, and the U.S. National Institutes of Health ACTIV and PETAL clinical trials networks; several authors (BW, IP, JB, SB, ES) were site investigators on these trials but received no direct or indirect remuneration for their effort. ES, BJW, SMB and MS are members of the Utah crisis standards of care scarce medication committee.

chemotherapy, biologic or other immunosuppressive agents targeting B or T cell activity, chronic corticosteroids at a prednisone-equivalent dose of 20mg per day or greater for more than 30 days, human immunodeficiency virus complicated by acquired immunodeficiency syndrome (AIDS), heritable immunodeficiency. We defined obesity as body mass index (BMI) of greater than or equal to 30 [12]. Symptom and demographic data were complete; comorbidity data were complete insofar as patients had prior encounters in the integrated health system. Mortality data was captured via an existing linkage to state death records.

We used a random number generator to divide the cohort into a 70% derivation cohort and 30% validation cohort. In the derivation cohort data, we fitted a multivariable logistic regression model for hospitalization within 14 days of testing, using clinical and demographic features. Predictors were prespecified before model development based on: clinical features that would be available at the time of testing for all ambulatory and emergency department patients regardless of testing venue (a criterion that precludes, for instance, laboratory data), biological plausibility of association with severity, and reproducibility in other studies in existing COVID-19 literature. We intentionally did not fit a model for mortality, but instead planned *a priori* to validate the ultimate model against that outcome. Model discrimination was evaluated using the area under the receiver-operator characteristic curve (AUROC) and model fit by evaluating $R^2$ using the Nagelkerke method because the Hosmer-Lemeshow goodness-of-fit is not valid in very large sample sizes [13,14].

We included patients who tested in the ambulatory setting as well as patients who tested positive in the emergency department to ensure that the score would be applicable in both environments. However, we recognized that some patients testing positive in the emergency department (ED) are then subsequently admitted. The decision to admit or not is not immediately known to emergency medicine providers who may still wish to use the score to stratify risk to aid in clinical decision making and selection of therapies. However, because patients who are admitted from the ED may have different characteristics than those tested in the ambulatory setting, we planned *a priori* to perform a sensitivity analysis by repeating the regression above after restricting the cohort to patients who were not admitted to the hospitalization at the time of their test.

We then adapted the original logistic regression model into a simple scoring tool by converting exponentiated β coefficients into weighted point assignments for each variable. We evaluated the test performance characteristics of this simplified clinical prediction tool in the derivation and validation cohorts using AUROC and by calculating the sensitivity, specificity, negative and positive predictive values across the range of scoring thresholds. To account for possible secular changes in patient distribution or variant epidemiology, we performed a temporally-independent internal validation of the scoring tool in a cohort comprised of all laboratory-confirmed COVID-19 patients in the IPOC database from November 1, 2020 to August 15, 2021.

This study was approved by the Intermountain Healthcare Institutional Review Board which granted a waiver of informed consent to use patient data collected and stored for public health purposes.

## Results

From March 25 through October 1, 2020, 22,816 patients had a positive PCR test for SARS-CoV-2. The mean age was 40 years (see Table 1); 11,424 (50.1%) patients were female and 8753 (43.9%) identified as a member of a community of color (either non-white race or Hispanic or Latinx ethnicity). Patients had on average one significant medical comorbidity. 1419 (6.2%) of patients were admitted; of these, 799 (3.6%) tested positive in the emergency

**Table 1. Patient characteristics, total and by derivation and validation cohort groups.**

| | ALL | DERIVATION | VALIDATION |
|---|---|---|---|
| | N (%) unless noted | N (%) unless noted | N (%) unless noted |
| All Patients | 22816 (100) | 16030 (70.3) | 6786 (29.7) |
| Male | 11392 (49.9) | 8005 (49.9 | 3387 (49.9) |
| Age, years (Mean, SD) | 40.4 (16.5) | 40.4 (16.5) | 40.2 (16.6) |
| Race | | | |
| American Indian or Alaska Native | 238 (1.0) | 169 (1.1) | 69 (1.0) |
| Asian | 349 (1.5) | 238 (1.5) | 111 (1.6) |
| Black or African American | 341 (1.5) | 233 (1.5) | 108 (1.6) |
| Multiple | 78 (0.3) | 56 (0.3) | 22 (0.3) |
| Native Hawaiian or Pacific Islander | 893 (3.9) | 626 (3.9) | 267 (3.9) |
| White | 16624 (72.9) | 11637 (72.6) | 4987 (73.5) |
| Ethnicity | | | |
| Hispanic, Latino, or Spanish origin | 7027 (30.8) | 4980 (31.1) | 2047 (30.2) |
| Communities of Color[1] | 8753 (43.9) | 6184 (44.3) | 2569 (43.0) |
| Symptoms (Reported at time of test) | | | |
| Fever | 7889 (34.6) | 5561 (34.7) | 2328 (34.3) |
| Cough | 11595 (50.8) | 8188 (51.1) | 3407 (50.2) |
| Dyspnea | 6008 (26.3) | 4273 (26.7) | 1735 (25.6) |
| Myalgia | 11341 (49.7) | 7985 (49.8) | 3356 (49.5) |
| Rhinorrhea | 8843 (38.8) | 6203 (38.7) | 2640 (38.9) |
| Anosmia | 5164 (22.6) | 3681 (23.0) | 1483 (21.9) |
| Pharyngitis | 8130 (35.6) | 5718 (35.7) | 2412 (35.5) |
| Diarrhea | 3648 (16.0) | 2573 (16.1) | 1075 (15.8) |
| Comorbidities | | | |
| Count, Mean (SD), Range | 0.7 (1.3), 0–11 | 0.7 (1.3), 0–11 | 0.7 (1.3), 0–10 |
| Diabetes Mellitus | 2164 (9.5) | 1532 (9.6) | 632 (9.3) |
| Hypertension | 3897 (17.1) | 2816 (17.6) | 1081 (15.9) |
| Cardiovascular Disease | 331 (1.5) | 246 (1.5) | 85 (1.3) |
| Cardiac Arrhythmia | 2437 (10.7) | 1704 (10.6) | 733 (10.8) |
| Chronic Pulmonary Disease | 4231 (18.5) | 2920 (18.2) | 1311 (19.3) |
| Chronic Kidney Disease | 687 (3.0) | 507 (3.2) | 180 (2.7) |
| Congestive Heart Failure | 536 (2.3) | 384 (2.4) | 152 (2.2) |
| Chronic Liver Disease | 1320 (5.8) | 914 (5.7) | 406 (6.0) |
| Obesity | 3395 (14.9) | 2376 (14.8) | 1019 (15.0) |
| Immunosuppression | 143 (0.6) | 101 (0.6) | 42 (0.6) |
| Cerebrovascular Disease | 589 (2.6) | 409 (2.6) | 180 (2.7) |
| Neurological Disorders | 1037 (4.5) | 723 (4.5) | 314 (4.6) |
| History of Tobacco Use | 3324 (21.5) | 2295 (21.2) | 1029 (22.1) |
| Mortality, 28-Day All-Cause | 93 (0.4) | 73 (0.5) | 20 (0.3) |
| Hospitalization, 14-Day | 1419 (6.2) | 990 (6.2) | 429 (6.3) |

Abbreviations: SE: Standard Error.

[1]Self-identifies as either non-white race or Hispanic/Latinx ethnicity.

department during the encounter that culminated in admission. Overall 93 patients (0.4%) died within 28 days of their positive SARS-CoV-2 assay. Demographic and clinical features were very similar between derivation (n = 16,030) and hold-out validation (n = 6786) cohorts. Demographics for the temporally-independent validation cohort are presented in Table 2.

**Table 2. Demographics and clinical characteristics of the temporally-independent validation cohort.**

| | Laboratory-confirmed COVID-19 Positive Patients | |
|---|---|---|
| | N* | (%)* |
| Total, N (%) | 86,130 | |
| **Demographics** | | |
| Age, mean years (SD) | 42.5 | 16.9 |
| Female | 44892 | 52.1 |
| Race, American Indian or Alaskan Native | 662 | 0.8 |
| Race, Asian | 1072 | 1.2 |
| Race, Black or African American | 773 | 0.9 |
| Race, Native Hawaiian or Pacific Islander | 1416 | 1.6 |
| Race, White | 87525 | 82.2 |
| Race, other or multiple | 11378 | 13.2 |
| Hispanic or Latinx Ethnicity | 10859 | 12.6 |
| Community of Color | 23575 | 27.4 |
| **Symptoms (at time of positive test)** | | |
| Fever | 26197 | 30.4 |
| Cough | 45828 | 53.2 |
| Shortness of breath | 19251 | 22.4 |
| Myalgia | 44955 | 52.2 |
| Rhinorrhea | 38770 | 45.0 |
| Altered sense of smell | 18462 | 21.4 |
| Pharyngitis | 33095 | 38.4 |
| Diarrhea | 11886 | 13.8 |
| **Comorbidities** | | |
| Total Comorbidities, median (IQI) | 0 | 0–1 |
| Immunocompromised status | 603 | 0.7 |
| Diabetes Mellitus | 7186 | 8.3 |
| Coronary Artery Disease | 1408 | 1.6 |
| Active Malignancy | 560 | 0.7 |
| Chronic Pulmonary Disease | 19758 | 22.9 |
| Chronic Kidney Disease | 2976 | 3.5 |
| Chronic Liver Disease | 5484 | 6.4 |
| Cerebrovascular Disease | 2697 | 3.1 |
| Hypertension | 16724 | 19.4 |
| Chronic Neurological Disease | 4164 | 4.8 |
| Congestive Heart Failure | 2229 | 2.6 |
| Cardiac Arrhythmia | 10701 | 12.4 |
| Obesity | 14486 | 16.8 |
| **Outcomes** | | |
| Hospitalization within 14 days | 2555 | 3.0 |
| Mortality within 28 days | 293 | 0.3 |

In the derivation cohort, clinical features by hospitalization status are reported in Table 3. The primary multivariable model (see Table 4) demonstrated adequate model diagnostics [AUROC 0.824 (95% CI 0.809–0.840), Nagelkerke $R^2$ 0.26]. Age, male sex, self-identification to a community of color, dyspnea and high-risk comorbidities including diabetes mellitus, obesity, immunosuppression and chronic neurologic disease were each associated with

**Table 3. Patient characteristics of the derivation cohort stratified by outcome of hospitalization.**

| | Hospitalized | |
|---|---|---|
| | **No** | **Yes** |
| | N (%) unless noted | N (%) unless noted |
| **N =** | 15040 | 990 |
| **Male** | 7472 (49.7%) | 533 (53.8) |
| Age, Years (Mean, SD) | 39.5 (16) | 54.8 (17.7) |
| **Race** | | |
| American Indian or Alaska Native | 145 (1.0) | 24 (2.4) |
| Asian | 218 (1.4) | 20 (2.0) |
| Black or African American | 217 (1.4) | 16 (1.6) |
| Multiple | 55 (0.4) | 1 (0.1) |
| Native Hawaiian or Pacific Islander | 524 (3.5) | 102 (10.3) |
| White | 10940 (72.7) | 697 (70.4) |
| **Ethnicity** | | |
| Hispanic or Latinx | 4622 (30.7) | 358 (36.2) |
| Communities of Color[1] | 5671 (43.5) | 513 (54.7) |
| Symptoms (at time of testing) | | |
| Fever | 4999 (33.2) | 562 (56.8) |
| Cough | 7575 (50.4) | 613 (61.9) |
| Dyspnea | 3707 (24.6) | 566 (57.2) |
| Myalgia | 7462 (49.6) | 523 (52.8) |
| Rhinorrhea | 5961 (39.6) | 242 (24.4) |
| Anosmia | 3516 (23.4) | 165 (16.7) |
| Pharyngitis | 5476 (36.4) | 242 (24.4) |
| Diarrhea | 2379 (15.8) | 194 (19.6) |
| **Comorbidities** | | |
| Comorbidity Count, (Mean, SD), Range | 0.7 (1.2), 0–10 | 2.1 (2.0) 0–11 |
| Diabetes Mellitus | 1145 (7.6) | 387 (39.1) |
| Hypertension | 2308 (15.3) | 508 (51.3) |
| Cardiovascular Disease | 178 (1.2) | 68 (6.9) |
| Cardiac Arrhythmia | 1442 (9.6) | 262 (26.5) |
| Chronic Pulmonary Disease | 2620 (17.4) | 300 (30.3) |
| Chronic Kidney Disease | 357 (2.4) | 150 (15.2) |
| Congestive Heart Failure | 261 (1.7) | 123 (12.4) |
| Chronic Liver Disease | 768 (5.1) | 146 (14.7) |
| Obesity | 1987 (13.2) | 389 (39.3) |
| Immunosuppression | 82 (0.5) | 19 (1.9) |
| Cerebrovascular Disease | 315 (2.1) | 94 (9.5) |
| Chronic Neurological Disease | 573 (3.8) | 150 (15.2) |
| **History of Tobacco Use** | 2012 (20.5) | 283 (28.6) |
| **Mortality, 28-Day All-Cause** | 13 (0.1) | 60 (6.1) |

Abbreviations: SE: Standard Error.

[1]Self-identifies as either non-white race or Hispanic/Latinx ethnicity.

significantly greater odds of hospitalization. In an exploratory analysis in which individual comorbidities were replaced in the regression with a count of total comorbidities, the cumulative number of comorbidities was also significant (OR 1.4, 95% CI 1.3–1.5). In the planned sensitivity analysis excluding patients who were tested in the emergency department during

**Table 4. Multivariable logistic regression model for hospitalization in the derivation cohort.**

|  | p | Adjusted Odds Ratio | 95% CI |
|---|---|---|---|
| Age (decades) | <0.0001 | 1.5 | 1.4–1.6 |
| Male | <0.0001 | 1.3 | 1.2–1.6 |
| Communities of color[1] | <0.0001 | 2.1 | 1.8–2.4 |
| Dyspnea | <0.0001 | 3.5 | 3.0–4.0 |
| Diabetes mellitus | <0.0001 | 2.2 | 1.8–2.6 |
| Hypertension | 0.001 | 1.4 | 1.1–1.7 |
| Coronary Artery Disease | 0.45 | 0.88 | 0.61–1.3 |
| Cardiac Arrhythmia | 0.41 | 1.1 | 0.9–1.3 |
| Chronic Pulmonary Disease | 0.39 | 0.92 | 0.8–1.1 |
| Chronic Kidney Disease | 0.29 | 1.1 | 0.9–1.5 |
| Congestive Heart Failure | 0.07 | 1.3 | 1.0–1.8 |
| Chronic Liver Disease | 0.98 | 1.0 | 0.8–1.2 |
| Obesity | <0.0001 | 1.9 | 1.6–2.3 |
| Immunosuppression[2] | 0.02 | 3.9 | 1.3–12.1 |
| Cerebrovascular Disease | 0.74 | 1.1 | 0.8–1.4 |
| Chronic Neurologic Disease | <0.0001 | 1.8 | 1.4–2.4 |

[1]Self-identifies as either non-white race or Hispanic/Latinx ethnicity.

[2]Excludes patients with metastatic cancer with non-hospitalization goals of care.

their admission to the hospital, the multivariable model had slightly diminished performance [AUROC 0.789 (95% CI: 0.768–0.810), $R^2$ 0.164]. Overall, contributions of individual risk factors were similar in this model compared to the model including patients being admitted, (see Table 5) with the exception that the magnitude of risk of dyspnea was less in the ambulatory-

**Table 5. Sensitivity Analysis: Multivariable logistic regression model for hospitalization in the derivation cohort, excluding patients admitted from the emergency department.**

|  | p | Adjusted Odds Ratio | 95% CI |
|---|---|---|---|
| Age (decades) | <0.0001 | 1.5 | 1.4–1.6 |
| Male | 0.003 | 1.3 | 1.1–1.6 |
| Communities of color[1] | <0.0001 | 1.8 | 1.5–2.2 |
| Dyspnea | <0.0001 | 2.1 | 1.7–2.5 |
| Diabetes mellitus | <0.0001 | 2.1 | 1.6–2.6 |
| Hypertension | 0.001 | 1.2 | 1.0–1.6 |
| Coronary Artery Disease | 0.91 | 1.0 | 0.6–1.5 |
| Cardiac Arrhythmia | 0.39 | 1.1 | 0.9–1.4 |
| Chronic Pulmonary Disease | 0.12 | 1.2 | 1.0–1.4 |
| Chronic Kidney Disease | 0.89 | 1.0 | 0.7–1.4 |
| Congestive Heart Failure | 0.72 | 1.1 | 0.7–1.6 |
| Chronic Liver Disease | 0.87 | 1.0 | 0.8–1.4 |
| Obesity | <0.0001 | 1.8 | 1.5–2.3 |
| Immunosuppression[2] | 0.003 | 7.0 | 2.0–24.9 |
| Cerebrovascular Disease | 0.25 | 1.2 | 0.7–1.5 |
| Chronic Neurologic Disease | 0.81 | 1.0 | 1.4–2.4 |

[1]Self-identifies as either non-white race or Hispanic/Latinx ethnicity.

[2]Excludes patients with metastatic cancer with non-hospitalization goals of care.

**Table 6. Simplified clinical prediction score for COVID-19 outcomes.**

| Demographic Risk Factors | Points |
|---|---|
| Male | 1 |
| Age | **0.5 for every decade:**<br>0–10 = **0.5**, 11–20 = **1**, 21–30 = **1.5**, 31–40 = **2**, 41–50 = **2.5**, 51–60 = **3**, 61–70 = **3.5**, 71–80 = **4**, 81–90 = **4.5**, 91–100 = **5**, >100 = **5.5** |
| Communities of color[1] | 2 |
| **High Risk Comorbidities** | |
| Diabetes Mellitus | 2 |
| Severely Immunocompromised[2] | 2 |
| Obesity (BMI>30) | 2 |
| **Other Comorbidities** | |
| Hypertension | 1 |
| Coronary Artery Disease | 1 |
| Cardiac Arrhythmia | 1 |
| Congestive Heart Failure | 1 |
| Chronic Kidney Disease | 1 |
| Chronic Pulmonary Disease | 1 |
| Chronic Liver Disease | 1 |
| Cerebrovascular Disease | 1 |
| Chronic Neurologic Disease | 1 |
| **Symptom Risk Factor** | |
| Dyspnea | 1 |

[1]Self-identifies as either non-white race or Hispanic/Latinx ethnicity.

[2]Solid Organ or Bone Marrow Transplant, AIDS, Active Chemotherapy, or Inherited Immunodeficiency.

only cohort (OR 2.1 vs 3.5), and the odds of immunosuppressed patients without palliative goals of care being admitted were greater (OR 7.0 vs 3.9).

Criteria included in the probabilistic, simplified clinical prediction score are displayed in Table 6. Because cumulative comorbidity count was significantly associated with poor outcomes, we included comorbidities in the simplified tool that were not individually associated with increased risk in the expanded logistic regression model. In the derivation cohort, the AUROC for the simplified clinical prediction score for 14-day hospitalization was 0.82 (95% CI: 0.81–0.84) and 0.8 (95% CI: 0.78–0.82) in the validation cohort. AUROC for 28-day all-cause mortality in the derivation cohort was 0.91 (95% CI: 0.83–0.94) and in the hold-out validation cohort 0.80 (95% CI: 0.69–0.9). In the temporally-independent validation cohort, AUROC for hospitalization was 0.76 (95% CI 0.75–0.77) and for mortality 0.9 (95% CI 0.88–0.91). The scoring threshold that optimized sensitivity and specificity (by Youden's index [15]) was 6 with test characteristics of 71.1% and 76.2% respectively (Table 7). By comparison, in the derivation cohort, the AUROC for Charlson comorbidity index for predicting hospitalization was 0.74 (95% CI 0.72–0.77) and for mortality 0.82 (95% CI 0.69–0.95).

## Discussion

Given recent straining hospital volumes and the emergence of promising but limited-availability outpatient therapies for COVID-19, methods are needed to identify patients with COVID-19 at highest risk of progression to severe disease, hospitalization and death. Here we describe

**Table 7. Risk Score test characteristics across thresholds.**

| Point Threshold | Sensitivity | Specificity | PPV | NPV | % of Positives |
|:---:|:---:|:---:|:---:|:---:|:---:|
| 3 | 95.0% | 28.5% | 7.5% | 98.9% | 72.8% |
| 4 | 89.1% | 45.7% | 9.3% | 98.5% | 56.3% |
| 5 | 80.6% | 62.8% | 12.1% | 98.1% | 39.8% |
| 6 | 71.1% | 76.2% | 16.6% | 97.5% | 26.7% |
| 7 | 60.9% | 84.1% | 20.6% | 97.0% | 18.7% |
| 8 | 51.4% | 89.2% | 24.4% | 96.4% | 13.4% |
| 9 | 41.4% | 92.8% | 28.2% | 95.9% | 9.4% |
| 10 | 32.3% | 95.2% | 31.7% | 95.4% | 6.5% |
| 11 | 25.0% | 97.0% | 36.1% | 94.9% | 4.4% |
| 12 | 17.4% | 98.1% | 38.5% | 94.6% | 2.9% |

a simple scoring model capable of accurately risk stratifying ambulatory and emergency department patients for COVID-19 for subsequent hospitalization and mortality.

One of the primary strengths of this model is the simplified and easily calculable score using features that are widely accessible. In particular, our score does not require laboratory studies, which are unavailable in the majority of ambulatory patients testing positive for SARS-CoV-2. While preserving discriminative value, this simple scoring system has potential to facilitate more widespread clinical application in settings lacking robust integration of informatics. The model was derived and validated in a very large and diverse population in the western United States and is based on risk factors for severe disease that are largely conserved across global populations, including age, male sex, overall comorbid burden, and shortness of breath at the time of risk stratification. These factors align closely with those included in models derived in other locations and populations [3,5–9,16,17].

For comparison, the Jehi model [6] was derived and validated in a cohort of 4536 patients in Ohio, USA, and included age, gender, race/ethnicity, income and housing density, smoking status, symptoms, a small set of comorbidities (obesity, asthma, diabetes, hypertension and immunosuppression), as well as laboratory data if available. In the internal validation cohort, AUROC for predicting hospitalization for this model was 0.81. The Wollenstein-Betech model [8] used data from 91,000 Mexican patients, and demonstrated and AUROC of 0.62 using age, gender, chronic renal insufficiency, diabetes, immunosuppression, COPD, obesity, hypertension, tobacco use, cardiovascular disease and asthma. The Dashti model [5] was derived and internally validated in a cohort of more than 12,000 patients in Massachusetts, USA. Using age, gender, race/ethnicity, smoking status and median household income, this score had an AUROC for hospitalization risk of 0.77. Finally, we also compared the performance of the Charlson comorbidity index [10] in our own data and found that it was not quite as discriminative for hospitalization, but equally accurate at predicting mortality.

Although race and ethnicity are often omitted from clinical prediction models to prevent illegal or unethical profiling behavior, the National Quality Forum recommended that when applications of risk prediction include patient selection for preventive or therapeutic modalities, omission of race or ethnicity can actually cause inequity in healthcare access and worsen outcomes disparity by underestimating risk using other demographic and clinical features alone [18]. In COVID-19, it is now well-recognized that significant outcome differences among communities of color exist with respect to severe illness and hospitalization [19] despite adjustment for age, gender and underlying medical conditions [5,6]. This remains poorly understood and may be due to social determinants of health, inadequate access to healthcare, or poorly-controlled co-morbidities. Because we anticipated application of this risk

stratification model to aid in allocating preventive therapies in COVID, we, like other published models [5,6], chose to include race and ethnicity in our score. In future work, more refined socioeconomic, cultural and healthcare access surrogates would be preferable alternatives.

When emergency use authorization (EUA) was granted by the United States Food and Drug Administration for monoclonal antibodies bamlanivimab and casirivimab/imdevimab for administration in non-hospitalized patients with early mild-moderate COVID-19, most states were experiencing peak community transmission, with thousands of new patients per day. It became clear that not only would the supply of drugs be inadequate initially to treat all patients qualifying under EUA criteria, but the capacity to administer infusions without compromising infection control in infusion sites would be even more limited. To address this limited resource situation, the Utah Crisis Standards of Care scarce medications committee was convened with the goal of equitably and efficiently matching available infusion capacity to patients at highest probability of hospitalization most likely to benefit. The simple scoring tool described herein was ultimately adopted because of the simplicity, widely accessible clinical features and validation in a large, representative local population. By regularly adjusting the eligibility criteria based on the risk score threshold that best calibrates current infusion capacity to the number of new cases in high-risk strata, this risk-targeted drug allocation strategy has provided an equitable and flexible means of drug delivery in the context of still-uncertain efficacy and limited resources.

Limitations of our study include the retrospective, observational design, and the possibility that comorbidity data may have been unavailable or out of date for some patients in the cohort who receive the majority of their medical care outside our integrated healthcare system. Although the large study population and inclusion of widely recognized features improves the likelihood of generalizability, this will need to be confirmed through external validation before adoption in other populations.

## Conclusion

In this large retrospective cohort study, we identified simple risk factors that can easily be calculated at the bedside without laboratory values to risk stratify COVID-positive individuals for risk of hospitalization and death. Applications include guiding allocation of therapies that are limited in availability. External validation is needed to confirm generalizability in diverse and geographically independent population.

## Supporting information

**S1 File.**
(XLSX)

## Author Contributions

**Conceptualization:** Brandon J. Webb, Nancy Grisel, Samuel M. Brown, Ithan D. Peltan, Emily S. Spivak, Joseph Bledsoe.

**Data curation:** Brandon J. Webb, Nancy Grisel.

**Formal analysis:** Brandon J. Webb, Samuel M. Brown, Ithan D. Peltan, Joseph Bledsoe.

**Investigation:** Brandon J. Webb.

**Methodology:** Brandon J. Webb, Samuel M. Brown, Ithan D. Peltan, Joseph Bledsoe.

**Project administration:** Brandon J. Webb.

**Supervision:** Brandon J. Webb.

**Writing – original draft:** Brandon J. Webb, Nicholas M. Levin, Nancy Grisel, Samuel M. Brown, Ithan D. Peltan, Emily S. Spivak, Mark Shah, Eddie Stenehjem, Joseph Bledsoe.

**Writing – review & editing:** Brandon J. Webb, Nicholas M. Levin, Nancy Grisel, Samuel M. Brown, Ithan D. Peltan, Emily S. Spivak, Mark Shah, Eddie Stenehjem, Joseph Bledsoe.

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
