## [Decision Letter · Decision Letter 0]

16 Jun 2021

PONE-D-21-06171

Simple Scoring Tool to Estimate Risk of Hospitalization and Mortality in Ambulatory and Emergency Department Patients with COVID-19

PLOS ONE

Dear Dr. Webb,

Thank you for submitting your manuscript to PLOS ONE. After careful consideration, we feel that it has merit but does not fully meet PLOS ONE’s publication criteria as it currently stands. Therefore, we invite you to submit a revised version of the manuscript that addresses the points raised during the review process.

Required for acceptance are: external validation, calibration of the statistical model applied and comparison with other models already published.

We look forward to receiving your revised manuscript.

Kind regards,

Carlo Torti

Academic Editor

PLOS ONE

Additional Editor Comments:

Unfortunately, your work lacks external validation and comparison with known and validated risk scores. For these reasons, it is not suitable for publication as it appears now.

Journal Requirements:

3. In the Methods section, please provide additional information regarding the methods used to systematically guide the scoring tool development.

"IP reports salary support through a grant from the National Institutes of Health (U.S.A). SB reports salary support from the U.S. NIH, Centers for Disease Control and the Department of Defense; he also reports receiving support for chairing a data and safety monitoring board for a respiratory failure trial sponsored by Hamilton, effort paid to Intermountain for steering committee work for Faron Pharmaceuticals and Sedana Pharmaceuticals for ARDS work, support from Janssen for Influenza research, and royalties for books on religion and ethics from Oxford University Press/Brigham Young University. BW reports partial salary support from a U.S. Federal grant from AHRQ.  ES receives partial salary support through grants from the Centers for Disease Control.

At the time of submission, Intermountain Healthcare and the University of Utah have participated in COVID-19 trials sponsored by: Abbvie, Genentech, Gilead, Regeneron, Roche, and the U.S. National Institutes of Health ACTIV and PETAL clinical trials networks; several authors (BW, IP, JB, SB, ES) were site investigators on these trials but received no direct or indirect remuneration for their effort.  ES, BJW, SMB and MS are members of the Utah crisis standards of care scarce medication committee."

Reviewers' comments:

Reviewer's Responses to Questions

**Comments to the Author**

1. Is the manuscript technically sound, and do the data support the conclusions?

Reviewer #1: Yes

Reviewer #2: Yes

2. Has the statistical analysis been performed appropriately and rigorously? 

Reviewer #1: Yes

Reviewer #2: No

3. Have the authors made all data underlying the findings in their manuscript fully available?

Reviewer #1: Yes

Reviewer #2: No

4. Is the manuscript presented in an intelligible fashion and written in standard English?

Reviewer #1: Yes

Reviewer #2: Yes

5. Review Comments to the Author

Reviewer #1: The authors propose a new model capable of stratifying the risk of hospitalization and mortality in patients with COVID-19 outpatient or who access the emergency department.

The strengths of the score are:

1) the sample size and, consequently, the number of the score validation population;

2) the simplicity of the score, as it uses anamnestic and non-laboratory data.

Nonetheless, several issues need to be addressed by the authors:

1) Like all scores, it requires external validation, as the data is drawn from what is observed. For example, a state's health system, whether public or private, certainly influences health care towards the weakest and poorest social classes. This data can have repercussions on the results of the score. It is no coincidence that belonging to a more impoverished community represents a risk factor (OR 1.8, CI95% 1.5-2.2, p-value <0.0001).

2) A second observation regards the category of immunosuppressed patients. First of all, this label includes a multitude of different patients (recipient of a solid organ or hematopoietic stem cell transplant, on chemotherapy, biologic or other immunosuppressive agents targeting B or T cell activity, chronic corticosteroids at a prednisone-equivalent dose of 20 mg per day or more significant for more than 30 days, human immunodeficiency virus complicated by acquired immunodeficiency syndrome (AIDS), heritable immunodeficiency). It is hard to think that an HIV patient is similar to an immunodeficient patient because transplanted. Also, because the data in the literature show that some immunosuppressive states could even be protective. Furthermore, terminal metastatic patients not eligible for treatment have been excluded, and, consequently, we have no data on this population, which also falls into the category of patients considered "fragile". Being 6 months old doesn't make COVID infection any less dramatic. So, I would suggest breaking the "immunosuppression" label into its components and analyzed individually.

3) Finally, I have an objection to the usefulness of introducing a new score without making a comparison with those already present. For example, it would be nice to compare the data obtained against a known and validated score like the Charlson comorbidity index, which is similar to that proposed.

Reviewer #2: Simple Scoring Tool to Estimate Risk of Hospitalization and Mortality in Ambulatory and Emergency Department Patients with COVID-19

Title: Appropriate.

Abstract: Appropriate and informative.

Key words: Short 5 keywords are need; Hospitalization, Scoring, Cohort.

Introduction: Appropriate and informative.

Aim of work: Appropriate and informative.

Methodology: Appropriate and informative. IRB Approval/ Number is needed.

Results: Calibration of the final multivariate logistic regression model (The Hosmer–Lemeshow test) is needed.

The Discrimination of the final multivariate logistic regression model should be assessed by the area under the receiver operator characteristic (ROC) curve.

Discussion: Appropriate, however I expected more comparisons with recent studies on the same topic.

References: Please use Vancouver’s Style in all references.

6. PLOS authors have the option to publish the peer review history of their article (what does this mean?). If published, this will include your full peer review and any attached files.

Reviewer #1: No

Reviewer #2: No

---

## [Author Response · Author response to Decision Letter 0]

15 Nov 2021

Dear Dr. Torti,

Thank you for the thorough peer review or our original research article entitled “Simple Scoring Tool to Estimate Risk of Hospitalization and Mortality in Ambulatory and Emergency Department Patients with COVID-19” (PONE-D-21-06171). 

We have responded to the excellent suggestions by the reviewers. Please point-by-point table below. In response to comments by reviewers and editorial staff, have now conducted a robust independent validation in a very large, temporally-distinct cohort including more than 80,000 patients. This secondary validation cohort spans a different time period and addresses possible influence by changes in patient characteristics or the Delta variant. We have also included comparative validation of the Charlson Comorbidity index in our dataset and contrast our model with other published models in the discussion. 

We believe the paper is now significantly improved and propose that the current work be considered for publication with this validation basis. 

We have submitted a deidentified dataset with this revision.

Please see below for a revised conflicts of interest statement and author contributions.

Thank you for your consideration of this manuscript.

Sincerely,

Brandon J. Webb, MD, Corresponding author

Nicholas M. Levin, MD, Alternate Corresponding author

Brandon.Webb@imail.org

Nicholas.Levin@hsc.utah.edu

---

## [Decision Letter · Decision Letter 1]

6 Dec 2021

Simple Scoring Tool to Estimate Risk of Hospitalization and Mortality in Ambulatory and Emergency Department Patients with COVID-19

PONE-D-21-06171R1

Dear Dr. Webb,

We’re pleased to inform you that your manuscript has been judged scientifically suitable for publication and will be formally accepted for publication once it meets all outstanding technical requirements.

Kind regards,

Carlo Torti

Academic Editor

PLOS ONE

Additional Editor Comments (optional):

Reviewers' comments:

Reviewer's Responses to Questions

**Comments to the Author**

1. If the authors have adequately addressed your comments raised in a previous round of review and you feel that this manuscript is now acceptable for publication, you may indicate that here to bypass the “Comments to the Author” section, enter your conflict of interest statement in the “Confidential to Editor” section, and submit your "Accept" recommendation.

Reviewer #1: All comments have been addressed

2. Is the manuscript technically sound, and do the data support the conclusions?

Reviewer #1: Yes

3. Has the statistical analysis been performed appropriately and rigorously? 

Reviewer #1: Yes

4. Have the authors made all data underlying the findings in their manuscript fully available?

Reviewer #1: Yes

5. Is the manuscript presented in an intelligible fashion and written in standard English?

Reviewer #1: Yes

6. Review Comments to the Author

Reviewer #1: The Authors have completely addressed my concerns. The manuscript is significantly improved respect to the initial version. I have not further questions.

7. PLOS authors have the option to publish the peer review history of their article (what does this mean?). If published, this will include your full peer review and any attached files.

Reviewer #1: No

---

## [Editor Report · Acceptance letter]

18 Feb 2022

PONE-D-21-06171R1 

Simple Scoring Tool to Estimate Risk of Hospitalization and Mortality in Ambulatory and Emergency Department Patients with COVID-19 

Dear Dr. Webb:

I'm pleased to inform you that your manuscript has been deemed suitable for publication in PLOS ONE. Congratulations! Your manuscript is now with our production department. 

Kind regards, 

on behalf of

Dr. Carlo Torti 

Academic Editor

PLOS ONE